# Stratification, Scarification and Application of Phytohormones Promote Dormancy Breaking and Germination of Pelleted Scots Pine (*Pinus sylvestris* L.) Seeds

**Katarzyna Nawrot-Chorabik** [1,*], **Małgorzata Osmenda** [2], **Krzysztof Słowiński** [3], **Dariusz Latowski** [4], **Sylwester Tabor** [5] and **Stephen Woodward** [6]

1   Department of Forest Ecosystems Protection, Faculty of Forestry, University of Agriculture in Kraków, 29 Listopada Ave. 46, 31-425 Kraków, Poland

2   Olkusz Forest District, The State Forests National Forest Holding, 32-300 Olkusz, Poland; malgorzata.osmenda@katowice.lasy.gov.pl

3   Department of Forest Utilization, Engineering and Forest Technology, University of Agriculture in Kraków, 29-Listopada Ave. 46, 31-425 Kraków, Poland; krzysztof.slowinski@urk.edu.pl

4   Department of Plant Physiology and Biochemistry, Faculty of Biochemistry, Biophysics and Biotechnology, Jagiellonian University, Gronostajowa 7, 30-387 Kraków, Poland; dariusz.latowski@uj.edu.pl

5   Department of Production Engineering, Logistics and Applied Computer Science, Faculty of Production and Power Engineering, University of Agriculture in Kraków, Balicka 116b, 30-149 Kraków, Poland; sylwester.tabor@urk.edu.pl

6   School of Biological Sciences, University of Aberdeen, Cruickshank Building, St. Machar Drive, Aberdeen, Scotland AB24 3UU, UK; s.woodward@abdn.ac.uk

*   Correspondence: k.nawrot-chorabik@urk.edu.pl

**Abstract:** Seed quality is an important issue in forestry as it is an essential parameter in the production of high quality planting material. Many factors may hinder the harvesting of high quality seeds, including an insufficient number of sunny days, external conditions in temperate climate zones, and fungal pathogens affecting development of seedlings. We undertook to develop a procedure maximizing seed protection and promoting the optimum physiological development of seedlings by examination of the impact of seed pelleting (a general seed protection method) on germination rates and seedling development of Scots pine (*Pinus sylvestris* L.). Germination of pelleted seeds was examined in relation to substrate (water vs. soil) and LED light spectrum (white vs. red-blue). Several dormancy breaking treatments were applied: stratification/scarification, and growth regulator treatments including gibberellic acid (GA₃), indole-3-acetic acid (IAA) and 1-naphthaleneacetic acid (NAA), to stimulate seed germination. Experiments included independent tests for each treatment (stratification/scarification and plant growth regulators), and combinations of both stratification/scarification and phytohormone treatments. The impacts of these treatments and various controlled germination conditions on the fluorescence of chlorophyll were analyzed using the maximum efficiency of photosystem II photochemistry parameter (Fv/Fm). In addition, chlorophyll a and b content in Scots pine seedlings germinated from pelleted seeds, were quantified using high-performance liquid chromatography (HPLC). The combined stratification/scarification and growth regulator treatment was the most effective germination promoting method for pelleted Scots pine seeds. Scots pine seeds are highly likely to be photoblastic. The best germination rate, while maintaining optimal physiological parameters, was achieved in acidic soil (pH 5.0) with white LED light.

**Keywords:** germination; LED light; photoblastic seeds; phytochromes; seed pelleting

## 1. Introduction

Seed quality is an essential factor in the production of high quality planting material. The germination rate of seeds and development of viable seedlings can be hindered by

external conditions during the growing season in Poland and the occurrence of damping-off pathogens, including *Pythium*, *Phytophthora* and *Fusarium* spp. This adverse impact of fungus-like and fungal pathogens, and environmental conditions was the main reason for the development of seed pelleting in trees. Pelleting is a seed enhancement method in which intact seed is covered with a uniform layer of coating material composed of inert mineral filler and a binder (water or other adhesive), and enables the application of high doses of fungicides in the coating material [1]. In contrast, the lower germination rates of pelleted seeds, as compared to untreated ones, is a drawback. Another major problem, especially for conifer species, is seed dormancy, meaning that a proportion of viable seeds do not germinate even under favorable environmental conditions. The adaptive advantage of seed dormancy in natural conditions pertains to the lower probability of regeneration failure due to unfavorable environmental conditions, as not all seeds of a given generation germinate at the same time. However, this physiological property of seeds can result in uneven germination in forest nurseries and cause problems in the consistent production of high quality planting material [2]. These failures have been a problem for many years. Therefore, many scientists have started research on developing an effective method of breaking seed dormancy under controlled conditions.

In this work we tested various methods to break dormancy and promote germination under controlled lighting conditions, to maximize the germination rate of pelleted seeds and to obtain high quality seedlings free of pathogens and physiological disorders. One of the methods tested was the universal pre-sowing technique of stratification, used to break seed dormancy and to maximize germination rates in many plant species [3]. Various processes favor germination. It has been proven that treating seeds with low temperatures in the range of 4–6 °C for up to three months has a positive effect on the stimulation of enzymatic activity [4].

Seed scarification, another germination promoting method used for pelleted pine seeds, involves intentional damage to the impervious seed coat, enabling more rapid and even water uptake and gas exchange to the embryo [5]. Germination is also controlled by plant growth regulators such as auxins and gibberellins. Numerous studies [6] indicate that the auxin indole-3-acetic acid (IAA) has a positive effect on the germination rate of Scots pine seeds. Gibberellins, such as gibberellic acid $GA_3$, also promote germination of many tree species including Douglas fir (*Pseudotsuga menziesii* Mirb.) or Norway maple (*Acer platanoides* L.) [6–8]. The synthetic auxin, 1-naphthaleneacetic acid (NAA), was also investigated in studies. The 1-naphthaleneaetic acid (NAA) was used to promote growth of cells which build roots of the plant [9].

A recent improvement in plant tissue culture and seed germination is the use of LED light for artificial illumination. The advantage of LEDs is that the light spectrum to which the plants are exposed can be simply manipulated (e.g., red-blue or white) and adjusted to promote plant growth [10]. Wavelength affects numerous physiological processes including germination, vegetative growth (development of leaves and shoots) and generative growth (flowering) [11]. The photoblasty of pine seeds is unknown. General white light comprises mixed wavelengths in the visible spectrum, from 380–710 nm; its use in plant growth facilities generally provides the full spectrum required for growth. It is possible that germination of pine seeds is dependent on light received by phytochromes, the main light receptor in photomorphogenesis [12].

Investigations of signaling networks mediated by phytochromes and growth regulators have been performed on a model plant such as *Arabidopsis thaliana* (L.) Heynh.). This species is often chosen for experiments because of the well understood genome and the rapid growth of the plant. These criteria are not met by many plant species with very large genomes and slow growth rates. The conducted research suggested that germination of seed is regulated by both blue and red/far red light [12,13]. It appears that growth regulators such as $GA_3$ or abscisic acid mediate the signaling cascade in red/far red reception. Concentrations of $GA_3$, the main growth regulator involved in dormancy breaking, rise after exposure to red light in *A. thaliana* (L.) Heynh.) [13].

The vitality and physiological condition of seedlings, and thus, the quality of resulting planting material, can be estimated using a variety of physiological and biochemical parameters. For example, chlorophyll fluorescence can be used to estimate the impact of particular conditions on the level of stress sustained by seedlings [14].

The aim of the work reported here was to evaluate under controlled conditions, and in relation to substrate type and light spectrum, the impact of dormancy breaking and germination-promoting procedures on the germination of pelleted seeds of Scots pine, and the subsequent impacts of these treatments on the quality of the resulting seedlings. Seedling quality was assessed through measurements of the efficiency of photosystem II (Fv/Fm), and the concentrations of chlorophylls a and b [15].

## 2. Materials and Methods

### 2.1. Seed Material of Scots Pine and Germination Test Conditions

High quality seeds of Scots pine (*P. sylvestris* L.) were obtained from Jodłówka Forest Division (49°59′28″ N, 20°32′28″ E, Brzesko Forest District, Kraków Regional Directorate of National Forests Holding). Seeds were from a single stand and stored in refrigerated conditions (7 °C in the dark). For all experiments circa 1300 seeds were used.

The growth experiments were performed either in a germination room (phytotron, 23 °C +/− 1 °C) or in a Biogenet phytotron chamber equipped with Easy Green sprouters (Easy Green Factory, San Diego, CA, USA). The germination room was equipped with a separate ventilation system to minimize the risk of bacterial or fungal infections. Easy Green sprouters included separate rectangular trays (34 cm × 7.5 cm) and an internal misting system maintaining air humidity at 85.00%. Dedicated ventilated incubators (designed and built in the Department of Forest Utilization, Engineering and Forest Technology (DFUEFT), Faculty of Forestry, Hugo Kołłątaj University of Agriculture, Kraków), fitting two sprouters each, were used to ensure controlled light conditions. Incubators were equipped with independent LED light systems producing light over the optimal spectrum for plant growth and to maintain a fixed photoperiod. All experiments in sprouters were performed in three replications, with 50 seeds per replication, and a photoperiod of 12 h. Two different wavelengths were used in our experiments: combined red-blue, and white, in order to target different physiological processes. The red light (600–700 nm) is absorbed by chlorophyll a, whereas blue light (400–500 nm) modulates a variety of metabolic processes, including stimulating seed coat rupture and subsequent root and stem growth. White light (380–710 nm) was used as general photosynthetically active radiation (PAR) during the development of seedlings effecting, among others, the switch to autotrophy. Two different types of substrates were also used in the experiments: soil and distilled water. The soil was used as the primary substrate in which the seeds germinate and the seedlings grow in vivo. The second substrate used was distilled water. The moistened Scots pine seeds took up water and swelled, which initiated cracking of the seed coats and also initiated gas exchange.

Dormancy breaking comprised combined stratification and scarification performed with the following procedure: three 500 mL Simax bottles were each filled with 150 mL sterile coarse sand (particle size 0.4–0.7 mm) and 1000 of Scots pine seeds. Next, the growth regulator solution (0.01 mg/dm$^{-3}$ IAA, NAA and GA$_3$ was added, each to a different bottle). Bottles were stored for three weeks in the dark at 7 °C and shaken vigorously every two days to scarify the seeds. After this time seeds were removed from the gravel. For experiment variants involving only stratification/scarification or only growth regulators, no gravel or no growth regulator solutions were used, respectively.

Treated seeds were pelleted for 20 min with a dedicated drum-shaped rotary (electric powered) pelleting machine (designed and manufactured at DFUEFT); using mineral filler (silty-clay formations) previously ground with a pestle in a mortar and a water/gelatin binder (1:30). Pelleted seeds were dried at 21 °C until the coating solidified.

## 2.2. Tests Experimental Design

The impacts of the various described dormancy breaking procedures on the germination rate of pelleted seeds and on the functional development of Scots pine seedlings were tested under in vitro conditions. The experiment comprised five independent tests involving three dormancy breaking treatments (described above), and controls tests. Additionally, each test comprised two variants of substrate/light wavelength.

The control test comprised the germination of pelleted and non-pelleted seeds under natural daylight, to determine the baseline germination rate of both seed treatments without the application of dormancy breaking treatments. A total of 100 pine seeds (50 pelleted and 50 non-pelleted) were used in this treatment, sown into two pots filled with sterile soil. Pots were maintained in daylight at 24 °C and sprayed daily with water to ensure soil humidity around 80.00%. After three weeks, germination rates were determined for both control groups of seeds. Control tests demonstrated the seeds' germination rate in vivo.

The control tests were also carried out for pelleted and non-pelleted seeds sown on sterile soil, which were placed in pots and kept in darkness. A total of 100 pine seeds (50 pelleted and 50 non-pelleted) were used. The temperature in the darkroom was +/−24 °C. The humidity in the darkroom was about 70.00–80.00%. After three weeks, it was evaluated whether the pine seeds had germinated in the complete darkness.

Three dormancy breaking treatments used two substrates each (Table 1), soil pH 5.0 and deionized water pH 5.0. The seeds germinating in the soil were placed on the surface of the substrate. The seeds were slightly covered with soil. The seeds germinating in sprouters were moistened with distilled water. Similarly, two light treatments (described above) were used in the dormancy breaking tests: white and red-blue. No non-pelleted seeds were included in dormancy breaking tests, mostly due to the fact that standard procedures used for seed preparation in forest nurseries result in relatively good germination of high quality non-pelleted Scots pine seeds in daylight. Assessment: the germination rate of the uncoated seeds in vivo was assessed on the basis of control samples. The germination rate was determined 10 days after each treatment, and assessed as the mean number of viable seedlings emerging in three replicates of a given test (Table 1).

**Table 1.** Germination tests of pelleted *P. sylvestris* seeds in various substrate and light conditions. NA*—not applicable, IAA—indole-3-acetic acid, GA$_3$—gibberellic acid, NAA—1-naphthaleneacetic acid.

|   | Germination Test | Seed Coating | Dormancy Breaking Procedure | Growth Regulators (mg × dm$^{-3}$) | Substrate | Illumination Natural/No Light/ LED |
|---|------------------|--------------|-----------------------------|-------------------------------------|-----------|-------------------------------------|
| 1 | Control test, no dormancy breaking treatment | Pelleted Non-pelleted | NA* | NA* | Soil | Daylight/darkness |
| 2 | Control test, no dormancy breaking treatment | Pelleted | NA* | NA* | Water Soil | Red-blueWhite |
| 3 | Stratification/scarification | Pelleted | Stratification/ scarification | NA* | Water Soil | Red-blue White |
| 4 | Growth regulators | Pelleted | NA* | IAA 0.01 GA$_3$ 0.01 NAA 0.01 | Water Soil | Red-blue White |
| 5 | Stratification/scarification and growth regulators | Pelleted | Stratification/ scarification | IAA 0.01 GA$_3$ 0.01 NAA 0.01 | Water Soil | Red-blue White |

### 2.3. Chlorophyll Fluorescence in P. sylvestris Seedlings

The impact of the experimental treatments on photosynthetic efficiency in pine seedlings was examined using fluorescence methods, on seedlings from germination test 5. The seedlings (both: stratification/scarification and growth regulators, Table 1) were used 14 days after germination, and 12, vital, well-rooted Scots pine seedlings were acclimated in a Biogenet phytotron chamber. In the experiment, variable fluorescence (Fv) and maximum fluorescence (Fm) were measured using a hand-held PocketPEA (Hansatech Instruments, Norfolk, UK). These parameters were used to calculate the photosystem II photochemistry parameter (Fv/Fm). Measurements were carried out on each seedling placed in the dark for 20 min. Immediately after making measurements, seedlings were frozen in liquid nitrogen and stored at $-80$ °C until required for further analyses. An (Fv/Fm) value of 0.85 is the expected value for plants growing in optimal conditions (no stress), while (Fv/Fm) < 0.85 indicate the occurrence of stress resulting in reduced photosynthetic efficiency. Low (Fv/Fm) values (e.g., 0.2–0.3) indicate irreversible degradation of the photosystem II reaction centers.

### 2.4. Chlorophyll Extraction and Separation

Deep frozen seedlings from the previous analysis were homogenized in liquid nitrogen in a mortar and pestle and suspended in acetonitrile/methanol/water 82:8:1 (solvent A). The suspension was centrifuged ($16,900\times g$; 4 °C; 20 min), and the supernatant collected and stored on ice in the dark. Fresh solvent A (1 mL) was added to the pellet, vortexed and the centrifuging repeated. The procedure was repeated until the complete discoloration of the pellet. Pooled supernatants were analyzed on an Agilent 1260 Infinity HPLC system equipped with a photodiode array detector (Agilent Technologies, Waldbronn, Germany) on a Nucleosil 100 C18 column (5 μm 25 × 04) with solvent A as the mobile phase (0.8 mL/min). Chlorophylls a and b contents were quantified (absorbance at 440 nm) and applied to the appropriate spectrum of light length in the range from 380 nm to 710 nm with HPLC 1260 (offline): Data Analysis DryLab software, Molnár-Institute for applied chromatography, Berlin, Germany. Biochemical analyses were performed at the Faculty of Biochemistry, Biophysics and Biotechnology, Jagiellonian University.

### 2.5. Statistical Analyses

Results were analyzed using ANOVA, followed by a non-parametric NIR range test. The NIR test was used to identify significant differences ($\alpha = 0.05$) between treatments for groups of pelleted seeds germinating and growing under the various treatments. A series of four independent analyses was performed. Two tested for differences in quantities of seed germinating in regard to substrate (water vs. soil) and wavelength (white vs. red-blue). In the third analysis, differences between the levels of chlorophyll fluorescence resulting from the wavelength and from stimulation by particular growth regulators were examined. Differences in chlorophyll a and b content between seedlings growing on various substrates (water vs. soil) and light conditions (white vs. red-blue) were assessed in the fourth analysis. ANOVA (Fisher's test) proved that the assumption of uniformity amongst the examined groups was maintained and presented by the coefficient p–probability value.

## 3. Results

### 3.1. Germination of Seed

Rates of germination of seeds recorded in these control experiments varied based on treatment. Non-pelleted seed had a germination rate of 16.00% compared with that for pelleted seeds, 2.89% (Figure 1). The measurements showed a clear problem with the germination of enveloped or non-enveloped seeds growing in the soil under daylight conditions, without dormancy break treatments.

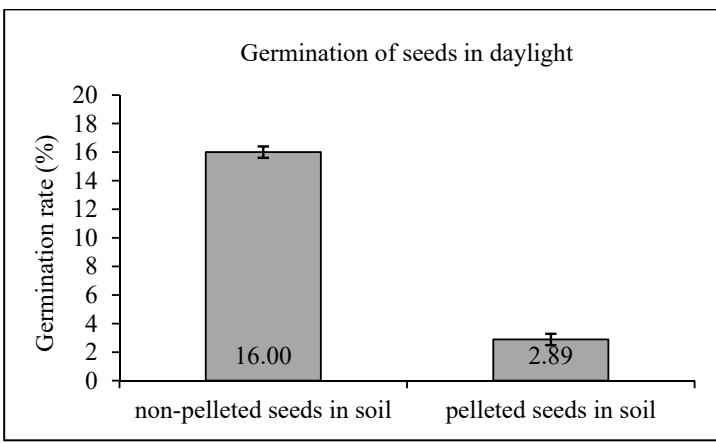

**Figure 1.** Germination rate in control experiment for pelleted and non-pelleted Scots pine seeds. Vertical bars indicate the standard deviation.

The use of additional light with different wavelengths, or the use of granulation, resulted in clear differences between the measurements. The low germination rate occurred when no additional treatment interrupting dormancy was performed (Table 2). Significant values for this test were observed in the soil and water substrate conditions. In soil, stratification/scarification of seed led to increased germination rates. In the case of pelleted seeds treated with stratification and scarification methods, the substrate type played a particularly important role in their germination, with high results observed with the water substrate. These differences were significant for seeds growing with previously presented combinations of factors (Table 3). The coated seeds that were in the soil substrate and in the complete darkness did not sprout at all (Table 2). The coated seeds treated with growth regulators did not obtain significant differences in different types of substrate and light (Table 3). Visible acceleration of germination rates of pelleted seeds occurred using a combination of growth regulators and stratification/scarification treatments (Table 2). These differences were significant for seeds grown in the presented conditions.

**Table 2.** Germination mean (%) of pelleted Scots pine seeds in relation to substrate, wavelength, and dormancy breaking treatment. \*\*GA$_3$—gibberellic acid 0.01 mg $\times$ dm$^{-3}$, NAA—1-naphthaleneacetic acid 0.01 mg $\times$ dm$^{-3}$, IAA—indole-3-acetic acid 0.01 mg $\times$ dm$^{-3}$.

| Substrate and Wavelength/Darkness | Control, No Dormancy Breaking Treatment (%) | Stratification/ Scarification (%) | Growth Regulators (%) | Stratification/Scarification and Growth Regulators (%) |
|---|---|---|---|---|
| Soil, white light | 4.00 | 12.00 | \*\*GA$_3$ 26<br>\*\*NAA 16<br>\*\*IAA 16<br>19.33 | \*\*GA$_3$ 93<br>\*\*NAA 60<br>\*\*IAA 35<br>62.67 |
| Soil, red-blue light | 4.00 | 12.00 | \*\*GA$_3$ 24<br>\*\*NAA 14<br>\*\*IAA 10<br>16.00 | \*\*GA$_3$ 80<br>\*\*NAA 52<br>\*\*IAA 40<br>57.33 |
| Soil, darkness | 0.00 | 0.00 | 0.00 | 0.00 |
| Water, white light | 6.67 | 23.33 | \*\*GA$_3$ 10<br>\*\*NAA 8<br>\*\*IAA 6<br>8.00 | \*\*GA$_3$ 85<br>\*\*NAA 70<br>\*\*IAA 61<br>72.00 |
| Water, red-blue light | 9.33 | 33.33 | \*\*GA$_3$ 16<br>\*\*NAA 12<br>\*\*IAA 10<br>12.67 | \*\*GA$_3$ 70<br>\*\*NAA 56<br>\*\*IAA 48<br>58.00 |

**Table 3.** Summary of statistical analyses of relationships between selected dormancy breaking methods, substrate types used for germination (soil vs. water), and supplementary light wavelengths (white vs. red-blue). NIR's multiple comparison test and ANOVA (Fisher's test). * *p*—probability value. * Significant differences (α = 0.05), values greater than the selected significance level are not important.

| | NIR's Multiple Comparison Test. | | | | Fisher's Test for Substrate Types | Fisher's Test for Light Colors |
|---|---|---|---|---|---|---|
| Dormancy breaking treatment | Soil substrate | Water substrate | White light | Red-blue light | | |
| Control | 0.0228 | 0.0332 | 0.136 | 0.111 | | |
| Stratification/scarification | 0.0035 | 0.0476 | 0.276 | 0.349 | * *p* = 0.84 | * *p* = 0.61 |
| Growth regulators | 0.0639 | 0.732 | 0.578 | 0.384 | | |
| Stratification/scarification and growth regulators | 0.000015 | 0.001014 | 0.0007 | 0.0052 | | |

### 3.2. Chlorophyll Fluorescence and Content

Among seedlings germinating under white light, the values of the photosystem II (Fv/Fm) photochemistry parameter were high (optimal values equal to 0.85 or close to the optimal value) in the soil substrate (Figure 2). The optimal value was found for seeds treated with $GA_3$ and slightly lower values of 0.81 for seeds treated with IAA and NAA. The (Fv/Fm) value was 0.83 for $GA_3$ treated seed germinated in water. The seeds treated with $GA_3$ and germinating in the soil and water had the best results from all the presented trials (Figure 2). The IAA and NAA treatments combined with sprouting in water gave lower (Fv/Fm) values, 0.61 and 0.35, respectively (Figure 2). Significant statistical differences were found for $GA_3$ treated pine seeds that germinated in deep soil and white light. For the remaining samples that germinated in white light, no significant statistical results were found (Table 3).

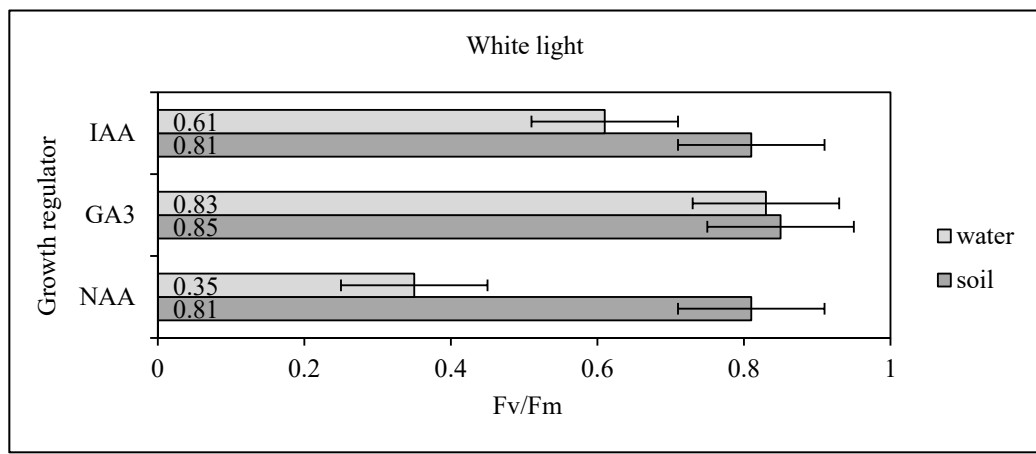

**Figure 2.** Efficiency of photosystem II photochemistry (Fv/Fm) for Scots pine seedlings germinated in white light. Vertical bars indicate the standard deviation. IAA—indole-3-acetic acid, $GA_3$—gibberellic acid, NAA—1-naphthaleneacetic acid.

Among seedlings germinating under red-blue light, the values of the photosystem II (Fv/Fm) photochemistry parameter were close to optimal when germinated in soil (Figure 3). The values were near optimal for seeds treated with all types of regulators (IAA, $GA_3$, NAA). Values estimated were between 0.84 and 0.81 (Figure 3). The best results for seeds germinating in the soil were those that were treated with NAA and $GA_3$. For seeds treated with $GA_3$, the value of the factor (Fv/Fm) was 0.81 while for the sample in which NAA was used, the value was 0.84. (Figure 3). The (Fv/Fm) factor values for the

seedlings germinated in water were lower than values for seeds that were growing in soil. The highest value for seeds germinating in water was found for seeds treated with $GA_3$ (0.78). The values for the seeds germinating in different types of growth regulators and in red-blue light did not show any significant statistical values (Table 5).

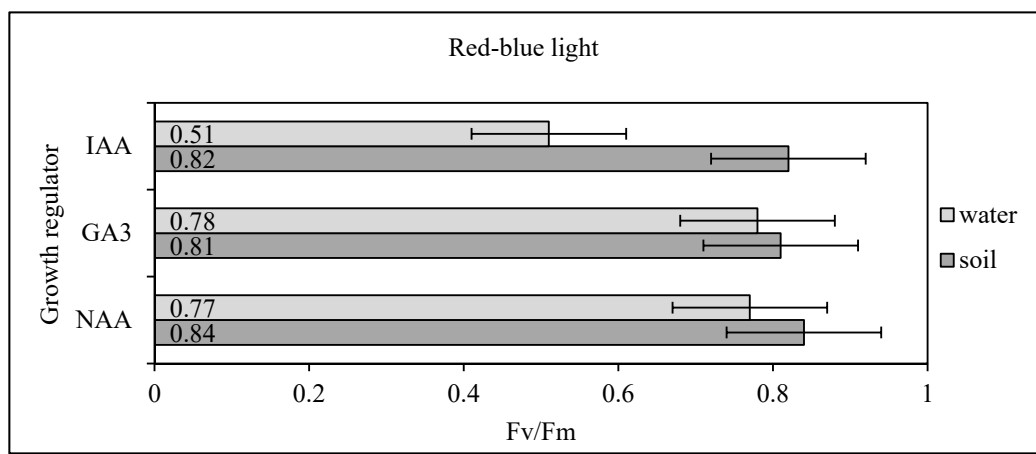

**Figure 3.** Efficiency of photosystem II photochemistry (Fv/Fm) for seedlings germinated in red-blue light. Vertical bars indicate the standard deviation. IAA—indole-3-acetic acid, $GA_3$—gibberellic acid, NAA—1-naphthaleneacetic acid.

### 3.3. Chlorophyll Concentrations

Highest concentrations of chlorophyll a to b ratios occurred in seedlings growing in soil with white light, regardless of the growth regulator used for breaking dormancy (Figure 4). Among these, the highest ratio of 1.859 occurred in seedlings from the $GA_3$ treatment. Lower chlorophyll a to b ratios occurred in red-blue, when seeds were germinated in water. Once again, the $GA_3$ treatment resulted in the highest ratios, compared with the two other growth regulator treatments. Seedlings germinated under red-blue light were characterized by lower chlorophyll a to b ratios in both substrates used for germination and all three growth regulator treatments. The lowest ratio was recorded for the combination: red-blue light, water and $GA_3$ treatment, whereas the NAA treatment resulted in the smallest differences in relative contents of chlorophyll a to b between substrate and wavelengths. Based on the values shown in Figure 4 we calculated the percentage ratio of the amount of chlorophyll a to b. These values emphasize the influence of external ones on the amount of vegetable dyes (Table 4). From these results, the differences in chlorophyll content proved to be statistically significant (Table 5). Statistical differences are noticeable in particular in chlorophyll a and in different conditions of the substrate and light, such as soil substrate and red-blue light, and water and white light. However, statistically significant differences in chlorophyll b did not appear in soil and white light and in water and red-blue light, while it was found in the combination of soil substrate and red-blue light, and water and white light (Table 6).

Table 4 shows the quantity of chlorophyll (a + b) in Scots pine seedlings in various substrate and irradiation treatments. The highest value was obtained for seedlings germinating in soil under white light, and the lowest, for seedlings germinating in water under red-blue light.

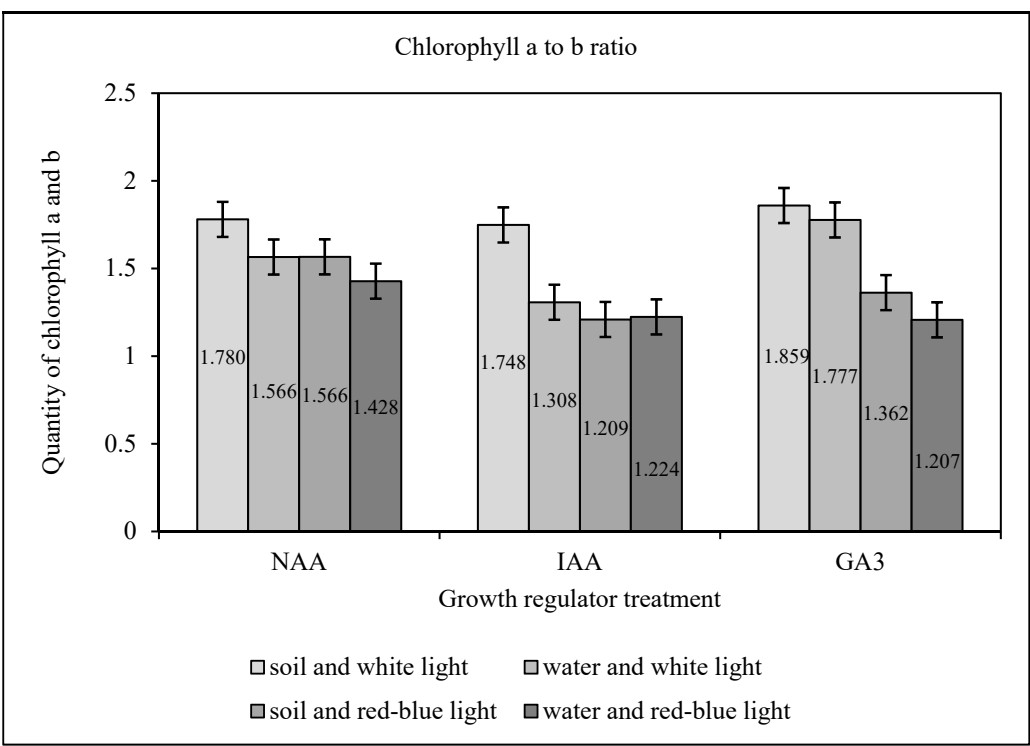

**Figure 4.** Chlorophyll a to b ratios in seedlings germinated from pelleted seeds under various controlled conditions. Vertical bars indicate the standard deviation. IAA—indole-3-acetic acid, GA3—gibberellic acid, NAA—1-naphthaleneacetic acid.

**Table 4.** Percentage amounts of chlorophyll in Scots pine seedlings germinated in various treatment conditions. * 100 percent of the largest ratio Chlorophyll a to chlorophyll b (Chla/Chlb) = 1.859 was the basis for calculating the percentages.

| Soil Type and Light Color | Mean Content of Chlorophyll a and b (%) * |
|---|---|
| soil and white light | 97.00 |
| water and white light | 83.00 |
| soil and red-blue light | 74.00 |
| water and red-blue light | 69.00 |

**Table 5.** Summary of statistical analysis of relationships between chlorophyll fluorescence, (Fv/Fm) light supplementary light treatments (white vs. red-blue), and growth regulator treatments (NAA, GA3 and IAA). NIR's multiple comparison test and ANOVA (Fisher's test). * $p$—probability value. * Significant differences ($\alpha$ = 0.05), values greater than the selected significance level are not important.

| NIR's Multiple Comparison Test | | | Fisher's Test for Light Colors |
|---|---|---|---|
| Growth regulator | White light | Red-blue light | |
| NAA | 0.273 | 0.490 | |
| GA3 | 0.028 | 0.122 | * $p$ = 0.20 |
| IAA | 0.221 | 0.371 | |

**Table 6.** Summary of statistical analysis of relationships between chlorophyll a and b ratios, substrate types (soil vs. water) and supplementary light treatments (white vs. red-blue). * *p*—probability value. * Significant differences ($\alpha = 0.05$), values greater than the selected significance level are not important.

| | NIR's Multiple Comparison Test | | Fisher's Test for Chlorophylls |
|---|---|---|---|
| Substrate type and wavelength | Chlorophyll a content | Chlorophyll b content | |
| Soil and white light | 0.101 | 0.163 | |
| Water and white light | 0.013 | 0.025 | * $p = 0.43$ |
| Soil and red-blue light | 0.0049 | 0.0079 | |
| Water and red-blue light | 0.082 | 0.083 | |

## 4. Discussion

The physiology of seed germination is regulated by multiple factors, beginning with imbibing water to activate the enzymatic machinery required for the metabolic reactions hydrolyzing storage materials used to synthesize new molecules and drive cell division and growth. These processes activate the embryos, which develop further and subsequently germinate. Many of the factors in this process, either internal (various metabolic processes) or external (light quality, oxygen, water or substrate proprieties), have long been known [16]. Seeds of many forest tree species are not easy to germinate, and enhancement methods for some of these species are not well developed. For example, the optimal external light conditions for germination, i.e., the photoblastic response, has not been determined for many species, including Scots pine. The results obtained in the present work indicated that Scots pine seeds are highly probable to be photoblastic and require at least 12 h of light (e.g., red-blue in the first germination phase and white later on). The development of seedlings from seeds characterized by their good quality, may be threatened by external abiotic and biotic factors. The abiotic factors that can threaten the germination process and seed's development, are radiation, and water shortage or excess. Seeds weakened by unfavorable abiotic factors may be susceptible to be attacked by pathogens such as fungi. For these reasons, effective seedling production techniques must be developed for forest nurseries. These techniques include methods of disrupting dormancy and promoting the healthy development of plants that result from germination.

The most important question is why should we apply seed pelleting and the interruption of seed dormancy on forest nurseries while standard procedures used for seed preparation in forest nurseries result in relatively good germination of non-pelleted Scots pine seeds. The answer to the question, is for the protection of seeds against external factors and the for the highest number of germinated seeds to obtain seedlings. An example of a situation to use these methods, is when seeds are of good quality and the amount of seed is extremely small. This can be due to a decrease in the condition of the best stands due to changes in environmental conditions such as temperature and water. These factors directly affect the quality and quantity of seeds. The results described in this manuscript demonstrated the effectiveness of dormancy breaking treatments in increasing germination rates of pelleted seeds and the promotion of growth and vitality in Scots pine. One of the techniques applied, stratification, is commonly used for breaking deep dormancy of seeds. The ease and effectiveness of this method sometimes combined with scarification, is routinely used for dormancy breaking in multiple conifer species [17]. Plant growth regulator treatments included in our research, in particular gibberellic acid (GA₃), also proved to be an effective tool in improving germination of Scots pine seeds.

Similar procedures are used in many economically important tree species, not only to stimulate germination, but also to improve health and vitality of the resulting seedlings. An example of such an approach was the application of these techniques to produce healthy planting material of *Pinus massoniana* (Lamb.), an important plantation species in Southeastern China, with high wood quality [18]. Low germination rates arose, probably due to stress induced by climate change, making it necessary to apply procedures stimulating germination and seedling development. Stratification and growth regulator treatments

increased both the number and quality of seedlings produced. In addition, GA and IAA were shown to enhance seed respiration rates, contributing to more rapid interruption of physiological dormancy [18].

The key factor that improved the germination of pelleted Scots pine seeds in the work reported here, was the combination of mechanical (stratification/scarification) and growth regulator treatments. Therefore, in the future, pelleting may prove an effective pre-treatment for the protection of plants against pathogens or bacteria. In recent years the presence of a species of fungal pathogen *Lecanosticta acicola* (Thümen) H. Sydow has been observed, which causes the disease called brown spot of pine needles [19]. This pathogen is an alien species which comes to Europe from Central America [20]. This disease affects pine species such as Scots pine and dwarf mountain pine (*Pinus mugo* Turra) [21]. On the basis of a number observations and studies, it was found that *P. mugo* (Turra) is very susceptible to infections by *L. acicula* (Thümen) which in Europe, puts populations at risk of strong damage in natural stands, or dye out [20]. On the basis of the presented research we can conclude, with coated seeds we can protect a lot of species of trees before damage, because the formulae of pelleting substances include fungicides that protect seeds and seedling from pathogens. This treatment is very important because fungicides included in the pelleted seeds protect the germinating seedling in the first stages of its development. This action minimizes the quick development of fungal disease in older seedlings when there is a big competition between seedlings. This topic is important because ecologically (*P. mugo* Turra) and economically (*P. sylvestris* L.) valuable tree species can be protected.

Pelleting can reduce the use of pesticides in plant production systems. Mechanical seed pelleting is a relatively simple and rapid procedure, resulting in a uniform and relatively thick seed coverage with a mineral coating that can also be supplemented with fungicides or other pesticides. Pelleting, therefore, ensures more targeted use of agrochemical products. Comparative studies in agriculture indicate that the introduction of seed pelleting reduced the requirement for chemical control agents by up to 85.00%, compared with traditional pesticide application methods [1]. Thus, the introduction of seed coating by pelleting with minerals will reduce the use of pesticides, with a consequent positive impact on and the environmental. Moreover, this process will reduce health risks to pesticide-handling personnel and the harmful effects of pesticide on soil organisms [1].

In the open environment, the main factors affecting seed germination are external conditions, such as light affecting seed germination intensity and quality, and substrate pro-prieties. Poland is located entirely in a temperate zone, but has wide ranging topography, including lowlands, highlands and mountain ranges, which impact on local temperatures, and the length of the vegetation season. Consequently, the quality of seedlings, the first and critical factor in regeneration of forest complexes, is equally varied. An alternative to growing seedlings in traditional open and container nurseries, may be the use of modern sprouters equipped with artificial LED light. LEDs have many advantages, positively affecting the development of seeds and plant tissues [11] and the economics of production. Specifically, these include the relatively small dimensions of individual fixtures, long lifespans [22], and increased safety of plant organs and tissues through lower intensity of light emitted by diodes (compared with mercury vapour lamps, for example) [23]. However, the biggest advantage of LED light is the ease with which the wavelength can be precisely adjusted for optimal plant metabolic response and development [24,25]. LED light is also cost-effective as it is significantly more energy efficient compared to other light sources [11].

One of the risks for seed germinating in natural conditions is light stress (photoinhibition) [26]. Chlorophyll fluorescence, estimated using (Fv/Fm) as well as by direct measurements of extracted photosynthetic pigments, can be used to determine if light stress is occurring [27] The photosynthetically active pigments, mainly chlorophyll a and b, are responsible for light absorption; lack or excess of light and/or an inadequate light spectrum may disrupt photosynthetic efficiency, eventually resulting with time in irreversible damage to the plant [28,29]. Therefore, monitoring of these parameters can be a useful tool in the development of controlled conditions aimed at minimizing the light

stress to seedlings of plants. The experiment carried out with the measurement of the photochemistry parameter of photosystem II (Fv/Fm) showed that the selection of an appropriate combination of light color and growth regulator, influences the physiological balance of the plant in controlled conditions. In particular, the combination of white light and $GA_3$ significantly influenced the optimal value achievement of the photochemistry parameter for photosystem II (Fv/Fm). (Figure 2, Table 5). In comparison, the conducted research showed that appropriate methods of interrupting dormancy affects the amount of chlorophyll necessary for photosynthesis in young seedlings (Figure 4 and Table 4).

Clearly, the dormancy breaking techniques described in this paper have great potential in forestry plant production. It should be emphasized, however, that the response to particular dormancy breaking treatments, coating methods, and germination conditions (light and substrate) vary significantly between plant species [30]. In addition, a certain level of variation in response to these stimuli is also expected due to phenotypic differences among provenances and individual plants. Nevertheless, the results obtained in this present work suggest that rapid progress could be made in the improvement of seed germination rates and production of vigorous seedlings for most important forest tree species used in Poland and elsewhere.

## 5. Conclusions

Environmental conditions such as the type of substrate and light, as well as the influence of fungal pathogens, have a strong influence on the germination of pelleted Scots pine. The results of our studies showed the following conclusions: It is highly probable that pine seeds are positively photoblastic and require a photoperiod including at least 12 h of light to germinate efficiently. The combination of stratification/scarification with growth regulators, particularly $GA_3$ and soil, were the most effective germination promoting treatments for pelleted Scots pine seeds. Seeds growing in soil after $GA_3$ treatment reached a frequency of 93%. This result is about 33% higher than the NAA and about 58% higher than the IAA. Germination rates after this treatment in soil with white light, were 16 times higher compared to seeds growing under the same conditions, without dormancy breaking treatments. Plant growth regulators had positive effects on seed germination in soil, with $GA_3$ being the best stimulator. Moreover, efficiency of photosystem II photochemistry in seedlings germinated after $GA_3$ treatment were close to optimal, suggesting that the treatment did not cause undue stress to seedlings and also had a positive effect on photosynthetic pigment contents. White light obtained highly promising results in germination of pelleted seeds. In this article, we analyzed for the first time all the above-mentioned treatments and the factors influencing the dormancy and germination of pine seeds.

**Author Contributions:** Conceptualization, K.N.-C.; methodology, K.N.-C. and D.L. and K.S.; software, M.O.; validation, S.T.; writing—original draft preparation, K.N.-C.; writing—review and editing, K.N.-C. and D.L. and M.O. and S.W.; visualization, K.N.-C. and D.L. and M.O.; supervision, K.N.-C. and S.W.; funding acquisition, K.S. All authors have read and agreed to the published version of the manuscript.

**Funding:** This work was from a Subvention of the Ministry of Science and Higher Education in Poland SUB/2019-0419 000 000-D404.

**Institutional Review Board Statement:** Not applicable.

**Informed Consent Statement:** Not applicable.

**Data Availability Statement:** Department of Forest Ecosystem Protection, University of Agriculture in Kraków, 29-Listopada Ave, 46, 31-425 Kraków, Poland; Department of Plant Physiology and Biochemistry, Faculty of Biochemistry, Biophysics and Biotechnology, Jagiellonian University, Gronostajowa 7, 30-387 Kraków, Poland; dariusz.latowski@uj.edu.pl.

**Conflicts of Interest:** The authors declare no conflict of interest.

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
