# Peer review of "Stratification, Scarification and Application of Phytohormones Promote Dormancy Breaking and Germination of Pelleted Scots Pine (Pinus sylvestris L.) Seeds"

_forests, doi:10.3390/f12050621_

Round 1

Reviewer 1 Report

There are numerous awkward sentances that should be reviewed- Lines 346,386,380,384,388,402,424,449 stand out in this regard.

Some other sssuggestions-

Ln 160 - I would like to see more detail regarding exactly what the water and soil treatments were. For example were the seeds floating in water? In the soil treatment, were the seeds covered with soil?

Ln 162-The authors state that standard procedures result in good germination %. In your article it would seem important to compare this standard procedure with the results of pelleting. At least give the reader a basis for comparison.

The authors make a number of claims about the role of pelleting conifer seeds in preventing a number of diseases. This study did not test that hypothesis yet makes a number of claims in that regard. For example, I find it questionable that seed coating with a fungicide will reduce brown spot fungus as the pine seedlings grow.  Be careful with this speculation.

Author Response

Dear Professor,

Thank you very much for all your comments as they were very helpful to me.

All the corrections in the manuscript were made using the track changes mode.

In addition, I also enclose the explanations for the following comments in Manuscript:

In the line 165 I explained that the seeds germinated in the sprouters did not float on the surface of the sprouters, but they were moistened with some distilled water. On the other hand, the seeds germinating in the soil substrate were sown on the surface of the substrate, where they were slightly covered with a layer of soil.

Secondly, the following corrections were made in relation to line 171:

The evaluation of the germination rate of non-pelleted seeds was presented in the control test. The outdoor conditions in the control test were similar to the in vivo conditions. The difference in the germination rate of non-pelleted and pelleted seeds in the control samples is presented in the figure 1.

In line 398 the role of seed coating with fungicide substrate is explained. Covering the seeds with a fungicide-containing substrate protects the seedling in the first stages of its growth. This treatment minimizes the risk of a strong fungal infection.

Explaining lines 440-445, the type of light in the extension they develop has an impact on the amount of chlorophyll and the contribution of germinating seedlings.

The explanation of the expression in lines 462–463 concerns the mean values obtained on the performed seed dormancy tests.

The minor remaining suggestions are explained in the manuscript.

I would like to thank you once again for all your amazing suggestions. Your contribution to my work is invaluable.

Reviewer 2 Report

Please find attached word document

Author Response

Dear Professor,

Thank you very much for all your comments as they were very helpful to me.

All the corrections in the manuscript were made using the track changes mode.

In addition, I also enclose the explanations for the following comments in Manuscript:

The photoblasticity of seeds results from the germination tests performed. If these seeds were physiologically weak, they were not photoblastic (line 36).

On line 38, the keywords were sorted alphabetically.

Following your hint, the title of subchapter 2.1 was changed (line 109).

Following your next suggestion, the wording in lines 117-120 has been moved to section 2.4 on chlorophyll extraction and separation methods. The wording can be found in lines 223-224.

The text of the manuscript explains the role of the substrates used, e.g. soil (lines 134-138).

As suggested, the title of subchapters 2.2 has been changed ( line 151).

Comments regarding uncoated seeds are explained in the manuscript. Non-pelleted seeds were tested as control sample (lines 157-164).

Suggestions regarding the temperatures at which the experiments were conducted can be found in the manuscript text (line 116 and 161).

In line 191, the age of the seedlings is specified.

Unnecessary text has been removed from the manuscript (197-202).

Removed the improperly bleached phrase found in verse 257-260.

Citations of the statistical tables (for example, Table 3) were not shown on the indicated line (for example, line 268) in the Manuscript as it does not comply with the "Forests" guidelines. Tables and figures should be placed after the first citation in the text. These changes proposed by the Reviewer would significantly disturb the structure of the Manuscript.

In the text of the manuscript there are values (Fv/Fm) for GA3-treated seeds in both types of areas (line 264-271).

After taking into consideration your suggestion, in line 297 white light was changed to red blue light. This is a very valuable comment - thank you.

Explanation of the expression in lines 375-376 in Table 2 presents methods of interrupting seed dormancy in controlled conditions, in which light conditions and the substrate play an important role. They influence the frequency of the seeds germination.

Seed coating can be one of the factors that reduces the use of pesticides. Pelleting, therefore, ensures more targeted use of products and reduces their use in the environment (lines 400-403).

Explaining lines 437-442, the type of light in the extension they develop has an impact on the amount of chlorophyll and the contribution of germinating seedlings.

After treating the seeds growing in the soil with gibelerin, 93% of them germinated, which gave correctly developed pine seedlings. This result (GA3) is 33% above the NAA and 58% above the IAA. These calculations are consistent with the given mean value in Table 2. Percentage results have been added to the text (lines 462-463).

The minor remaining suggestions are explained in the manuscript.

I would like to thank you once again for all your amazing suggestions. Your contribution to my work is invaluable.

Round 2

Reviewer 2 Report

Dear Authors,

Please find attached my suggestions

Author Response

Dear Professor,

Thank you very much for such a professional review. I have responded to all your comments and made an improvement. Now the Manuscript is much more understandable.
Attached I am enclosing a revised version of the Manuscript with a visible correction.

Again thank you very much.

Katarzyna Nawrot-Chorabik
